# Real-world comparative study of drug retention of Janus kinase inhibitors in patients with rheumatoid arthritis

Kenji Saito[1], Shuhei Yoshida[1], Honoka Ebina[1], Masayuki Miyata[2], Eiji Suzuki[3], Takashi Kanno[3], Yuya Sumichika[1], Haruki Matsumoto[1], Jumpei Temmoku[1], Yuya Fujita[1], Naoki Matsuoka[1], Tomoyuki Asano[1], Shuzo Sato[1], Kiyoshi Migita[1,4]*

1 Department of Rheumatology, Fukushima Medical University School of Medicine, Fukushima, Japan,
2 Department of Rheumatology, Japanese Red Cross Fukushima Hospital, Fukushima, Japan,
3 Department of Rheumatology, Ohta Nishinouchi General Hospital Foundation, Koriyama, Fukushima, Japan, 4 Department of Rheumatology, St. Francis Hospital, Nagasaki, Japan

* migita@fmu.ac.jp

**Data Availability Statement:** All relevant data are within the paper and its Supporting information files.

## Abstract

### Background

Janus kinase (JAK) inhibitors (JAKis) are effective therapeutic agents against rheumatoid arthritis (RA). However, patients having RA with particular risk factors may have a higher incidence of adverse effects (AEs), including major cardiovascular events (MACE) and infections. In this multicenter cohort study, we aimed to clarify the risk factors affecting the drug retention of JAKis in patients with RA.

### Methods

We retrospectively evaluated patients with RA who received their first JAKi (tofacitinib, baricitinib, upadacitinib, or filgotinib) at our institute. The clinical outcomes, including AEs, were recorded, particularly MACE and serious infections. The drug retention rates were analyzed using the Kaplan–Meier method, and risk factors affecting drug retention rates were determined using a multivariable Cox regression hazards model.

### Results

Overall 184 patients with RA receiving their first use of baricitinib (57.6%), tofacitinib (23.9%), upadacitinib (12.0%), or filgotinib (6.5%) were included in this study. Fifty-six (30.4%) patients discontinued JAKi treatment owing to ineffectiveness (9.2%) or AEs, including infections (21.2%). The overall drug retention rates were significantly lower in patients treated with pan-JAKi than in those treated with JAK1 inhibitors ($p = 0.03$). In the Cox regression model, the presence of baseline high RA disease activity, use of glucocorticoid and treatments with pan-JAKis were associated with reduced drug retention rates of JAKis ($p < 0.001$, $p = 0.01$ and 0.04, respectively). Pan-JAKi treated patients with high disease activity had significantly lower drug retention rates ($p < 0.001$).

**Funding:** The study was supported by the Japan Grant-in-Aid for Scientific Research (20K08777).

**Competing interests:** I have read the journal's policy and the authors of this manuscript have the following competing interests: K.M. has received research grants from Chugai Pharmaceutical Co., Ltd. and Novartis Pharma K.K. The above-mentioned pharmaceutical companies were not involved in the study design, data collection and analysis, manuscript writing, and manuscript submission. Rest of the authors declare that they have no conflict of interests.

## Conclusions

In a real-world setting, the drug retention rates of JAKis were reduced mainly by treatment discontinuation owing to AEs. Treatment with pan-JAKis and high baseline RA disease activity were identified as predictive factors for the discontinuation of JAKis. Lower drug retention rates were found in patients receiving pan-JAKis with high disease activity than in those without high disease activity.

## Introduction

Rheumatoid arthritis (RA) is a chronic autoimmune disease that causes synovial inflammation, leading to progressive joint destruction if optimal treatment is not provided [1]. The introduction of biological disease-modifying antirheumatic drugs (DMARDs) (bDMARDs) into conventional therapy has radically changed the prognosis of patients with RA [2]. However, a proportion of patients with RA remain refractory to these treatments [3]. Janus kinase (JAK) inhibition prevents the action of proinflammatory cytokines, such as interleukin-6 and granulocyte macrophage colony stimulating factor, by blocking the JAK/signal transducer and activator of transcription protein signal transduction pathway after the binding of cytokines to their receptors [4]. Recently, JAK inhibitors (JAKis) have been introduced in patients with RA who are refractory to conventional synthetic DMARDs or bDMARDs [5]. Indeed, JAKis tend to be introduced in patients with RA who inadequately respond to bDMARDs or methotrexate (MTX) in real-world settings [6].

Safety data concerning JAKis in clinical trials or their long-term extension studies have revealed an increased risk of infections, including herpes zoster [7]. However, in the real world, patients with RA may have higher risks for these adverse events (AEs) due to older age or the presence of comorbidities, both of which are exclusion criteria in clinical trials. Furthermore, a recent post-marketing safety survey, the ORAL surveillance study, raised concerns about an increased risk of malignancies and major adverse cardiovascular events associated with tofacitinib, compared with tumor necrosis factor inhibitors, in patients with RA with particular risk factors [8]. Currently, real-world evidence regarding the safety of JAKi is lacking. Therefore, it is important to identify factors that affect the effectiveness and safety of JAKis in patients with RA. In this study, we aimed to assess the clinical outcomes of JAKi therapy in a real-world population by evaluating treatment discontinuation owing to AEs or ineffectiveness. The secondary objective of this study was to identify the risk factors for drug discontinuation.

## Materials and methods

### Patients and study design

A multicenter retrospective cohort study was conducted to evaluate the risk factors affecting drug retention of JAKis in patients with RA. The study cohort consisted of patients treated at the department of rheumatology of Fukushima Medical University Hospital, Japanese Red Cross Fukushima Hospital, and Ohta Nishinouchi Hospital. Between April 2013 and October 2023, JAKi therapy was initiated in 211 patients with RA. Among these, 197 started receiving JAKi therapy in our institution, and 184 with sufficient clinical data were available were enrolled in this study. All patients were diagnosed with RA according to the 2010 American College of Rheumatology/European League Against Rheumatism classification criteria for RA

[9]. All records were accessed for research purposes between November 1, 2023 and November 30, 2023. The demographic data recorded at the start of each patient's JAKi treatment included age, sex, disease duration, rheumatoid factor, anti-citrullinated protein antibodies, history of bDMARD use, coexistence of diabetes mellitus or lung disease, and concomitant medication (s). RA disease activity was assessed using the disease activity score 28 using C-reactive protein (DAS28-CRP) [10]. Blood laboratory tests, medical histories, and clinical findings of patients with RA were collected by reviewing electronic medical records.

The JAKi-treated patients received baricitinib 2 mg (in patients with renal impairment) or 4 mg once daily, tofacitinib 5 mg twice or once daily (in patients with liver impairment), upadacitinib 7.5 mg (in patients with renal impairment) or 15 mg once daily, and filgotinib 100 mg (in patients with renal impairment) or 200 mg once daily. The study was approved by the institutional review boards of Fukushima Medical University (No. 2021–157), Japanese Red Cross Fukushima Hospital (No. 55), and Ohta Nishinouchi Hospital (No. 2022–8). Owing to the retrospective study design, an opt-out strategy was chosen for the partic-ipants, and those who declined to provide informed consent were excluded.

### Definitions of exposure and outcomes

Tofacitinib and baricitinib were defined as pan-JAKis, while upadacitinib and filgotinib as JAK1 selective inhibitors [11]. "Exposure" was defined as the period from the initiation of JAKi treatment until treatment discontinuation or death or the end of the study period, which-ever occurred first. "Discontinuation" was defined as discontinuation of treatment due to JAKi adverse events or a change in therapeutic agent due to ineffective treatment. The censoring time of the adverse events was defined as the time from the administration of the first dose of JAKi until the end of treatment or last observation point (October 31, 2023).

### Statistical analysis

Data are presented as median and interquartile range for continuous variables and frequency and percentage for qualitative variables. Mann–Whitney U test was used to compare continu-ous variables and Fisher's exact test to compare qualitative variables. Statistical significance was defined by two-tailed $p < 0.05$. The time to JAKi discontinuation in the treatment groups was estimated using Kaplan–Meier analysis, and log-rank tests were used to compare the cumulative IRs between the groups. Univariate and multivariable Cox regression analyses were performed to identify factors related to the incidence of JAKi discontinuation. Variables with $p < 0.2$ were included in the multivariable Cox regression analysis for the analysis. In cases where the number of variables that could be entered in the multivariate analysis was lim-ited by the number of outcomes, the more significant variables were adopted. Statistical analy-ses were performed using SPSS Statistics software (version 25.0; IBM Corp., Armonk, NY, USA) and R (version 4.1.2; R Foundation for Statistical Computing, Vienna, Austria, http://www.R-project.org/ [accessed June 23, 2023]).

### Results

#### Patients' baseline characteristics

Of the 211 patients with RA who started JAKi therapy between April 2013 and October 2023, 184 were enrolled in the study (Fig 1). The background characteristics of the patients in the JAKi-treated group are summarized in Table 1. The 184 patients in the JAKi-treated group included 106, 44, and 34 patients who received baricitinib, tofacitinib, and JAK1 inhibitors

## Flow chart showing selection of this study cohorts

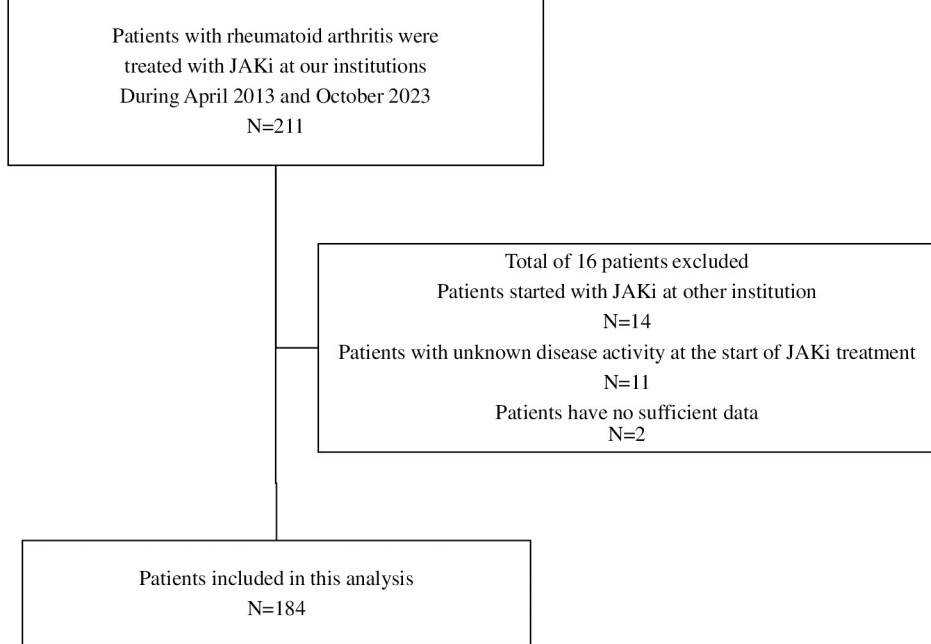

**Fig 1. Flow chart showing patient selection.** Among the 211 patients with RA who are initially treated with JAKis at our institution between April 2013 and October 2023, 184 for whom sufficient clinical data were available were enrolled in this study. RA: rheumatoid arthritis; JAKi: Janus kinase inhibitor.

(JAKis; upadacitinib and filgotinib), respectively. Of the 184 patients treated with JAKis, drug discontinuation occurred in 56 patients.

## Reasons for drug discontinuation

Overall, discontinuation of JAKis occurred in 56 patients, including 32, 22 and 2 treated with baricitinib, tofacitinib, and JAK1is, respectively (Table 2).

Among the 106 patients initiated with tofacitinib, drug discontinuation in 9 (8.5%) and 23 (21.7%) occurred owing to lack of effectiveness and AEs, respectively.

Among the 44 patients initiated with baricitinib, drug discontinuation in 7 (15.9%) and 15 (34.1%) occurred owing to lack of effectiveness and AEs, respectively.

Among the 34 patients initiated JAK1i, 1 patient each (2.9%) discontinued the drug due to lack of efficacy and AEs.

The details of the AEs are as follows (shown in Table 2): The AEs in patients initiated with baricitinib included bacterial pneumonia (n = 7), lymphoma (n = 4), soft tissue infections (n = 2), anemia (n = 2), lung cancer (n = 1), hepatocellular carcinoma (n = 1), pyothorax (n = 1), cholecystitis (n = 1), herpes zoster (n = 1), heart failure (n = 1), hepatic disorder (n = 1), and tinnitus (n = 1); the AEs in patients initiated with tofacitinib included bacterial pneumonia (n = 5), lymphoma (n = 2), cerebral hemorrhage (n = 2), colorectal cancer (n = 1), malignant melanoma (n = 1), lung cancer (n = 1), soft tissue infection (n = 1), leukopenia (n = 1), and exacerbation of interstitial pneumonia (n = 1); and the only AE observed among patients initiated with JAK1is was herpes zoster (n = 1).

**Table 1. Comparisons of clinical features in patients with RA treated with each JAKi.**

| | Baricitinib (n = 106) | Tofacitinib (n = 44) | JAK1i (n = 34) | |
| --- | --- | --- | --- | --- |
| | | | Upadacitinib (n = 22) | Filgotinib (n = 12) |
| Male, n (%) | 27 (25.5) | 12 (27.3) | 9 (40.9) | 6 (50.0) |
| Age at JAKi introduction, † years | 73 (65–83) | 72 (66–79) | 59 (56–66) | 66 (61–69) |
| Disease duration, † years | 7.9 (3.3–17.0) | 9.5 (4.1–15.4) | 4.6 (1.8–11.1) | 10.0 (6.0–21.8) |
| RF positivity, n (%) | 73 (68.9) no data 1 | 27 (61.4) | 16 (72.7) | 11 (91.7) |
| ACPA positivity, n (%) | 74 (69.8) no data 4 | 27 (61.4) no data 1 | 15 (68.2) no data 2 | 9 (75.0) no data 2 |
| Concomitant GC use, n (%) | 26 (24.5) | 12 (27.3) | 12 (54.5) | 6 (50.0) |
| Concomitant GC dose, † mg/day | 0.0 (0.0–0.0) | 0.0 (0.0–2.0) | 1.5 (0.0–2.5) | 5.0 (0.0–6.6) |
| Concomitant MTX use, n (%) | 39 (36.8) | 27 (61.4) | 10 (45.5) | 6 (50.0) |
| Concomitant MTX dose, † mg/week | 0.0 (0.0–6.0) | 5.0 (0.0–8.0) | 2.0 (0.0–8.0) | 4.0 (0.0–6.0) |
| Concomitant other DMARDs use, n (%) | 12 (11.3) | 4 (9.1) | 12 (54.5) | 3 (25.0) |
| Reduced dose of JAKi, n (%) | 60 (55.6) | 19 (43.2) | 7 (31.8) | 6 (50.0) |
| Coexisting ILD, n (%) | 16 (15.1) | 6 (13.6) | 2 (9.1) | 2 (16.7) |
| Coexisting DM, n (%) | 16 (15.1) | 9 (20.5) | 4 (18.2) | 1 (8.3) |
| DAS28-CRP at JAKi introduction, † | 3.3 (2.6–4.4) | 3.5 (2.7–3.8) | 2.9 (2.5–3.4) | 3.5 (2.9–3.8) |
| Reduced dose JAKi introduction, n (%) | 60 (56.6) | 19 (43.2) | 7 (31.8) | 6 (50.0) |
| Previous use of bDMARDs, n (%) | 50 (47.2) | 24 (54.5) | 14 (63.6) | 5 (41.7) |
| No. of previous use of bDMARDs, † | 1.0 (1.0–2.0) | 1.0 (0.0–1.0) | 2.0 (0.0–3.0) | 1.0 (0.0–2.3) |
| No. of DAS28-CRP >4.1 at JAKi introduction, n (%) | 32 (30.2) | 9 (20.5) | 3 (13.6) | 2 (16.7) |
| Observation period,† years | 1.8 (1.0–3.4) | 2.3 (1.0–3.7) | 1.9 (1.1–2.1) | 1.3 (0.8–1.5) |

†Values are the median with interquartile range. JAKi: janus kinase inhibitor, RA: rheumatoid arthritis, RF: rheumatoid factor, ACPA: anti-citrullinated peptide antibody, GC: glucocorticoid, MTX: methotrexate, DMARDs: disease-modifying anti-rheumatic drugs, ILD: interstitial lung disease, DM: diabetes mellitus, DAS28-CRP: disease activity score 28 using C-reactive protein, bDMARD: biologic disease-modifying anti-rheumatic drug.

### Factors associated with the discontinuation of JAKi treatment

Univariate and multivariable Cox regression analyses were performed to identify the predictors of overall JAKi discontinuation. glucocorticoid use, high baseline RA disease activity (DAS28-CRP > 4.1) and pan-JAKi use were identified as independent risk factors for the overall JAKi discontinuation (Table 3). Multivariable analysis was also performed to identify the risk factors for JAKi discontinuation owing to AEs. Baseline high RA disease activity (DAS28-CRP > 4.1) was identified an independent risk factor for drug discontinuation owing to AEs (Table 4).

**Table 2. AEs leading to JAKis discontinuation in patients treated with each JAKi.**

| Discontinuation of JAKi | Baricitinib (n = 23) | Tofacitinib (n = 15) | JAK1i (Upadacitinib+Filgotinib) (n = 1) |
| --- | --- | --- | --- |
| Infection, n (%) | 12 (52.2) | 6 (20.0) | 1 (100) |
| Malignancy, n (%) | 6 (26.1) | 5 (33.3) | 0 |
| Hematological disorder, n (%) | 2 (8.7) | 1 (6.7) | 0 |
| Cardiovascular disease, n (%) | 1 (4.3) | 2 (13.3) | 0 |
| Others, n (%) | 2 (8.7) | 1 (6.7) | 0 |

AEs; adverse events, JAKi: janus kinase inhibitor.

**Table 3. Independent risk factors of discontinuation of drug for all reasons in patients with RA treated with JAKi.**

| | Patients with RA treated with JAKi | | | |
|---|---|---|---|---|
| Variavle | Univariate model | | Multivariable model | |
| | HR (95% CI) | *p*-value | HR (95% CI) | *p*-value |
| Age, >75 yrs or not | 1.49 (0.87–2.55) | 0.15 | 1.14 (0.65–2.00) | 0.64 |
| Disease duration, per 1-yr increase | 1.00 (0.98–1.03) | 0.80 | | |
| Female or not | 0.79 (0.44–1.40) | 0.41 | | |
| RF positive or negative | 0.98 (0.56–1.73) | 0.95 | | |
| ACPA positive or negative | 1.13 (0.62–2.04) | 0.70 | | |
| GC use, yes/no | 1.49 (0.85–2.64) | 0.16 | 2.30 (1.27–4.18) | 0.01* |
| MTX use, yes/no | 0.77 (0.44–1.32) | 0.34 | | |
| Coexisting ILD, yes/no | 1.45 (0.71–2.98) | 0.31 | | |
| Coexisting DM, yes/no | 1.36 (0.70–2.63) | 0.36 | | |
| Pan-JAKi use, yes/no | 4.21 (1.01–17.45) | 0.048* | 4.49 (1.05–19.23) | 0.04* |
| DAS28-CRP >4.1 at JAKi introduction, yes/no | 3.04 (1.78–5.19) | <0.001* | 3.28 (1.86–5.77) | <0.001* |
| No. of previous use of bDMARDs, per drug | 1.09 (0.91–1.31) | 0.36 | | |
| Reduced dose of JAKi, yes/no | 0.82 (0.48–1.40) | 0.46 | | |

JAKi: janus kinase inhibitor, HR: hazard ratio, CI: confidence interval, RF: rheumatoid factor, ACPA: anti-citrullinated peptide antibody, GC: glucocorticoid, MTX: methotrexate, DMARDs: disease-modifying anti-rheumatic drugs, ILD: interstitial lung disease, DM: diabetes mellitus, DAS28-CRP: disease activity score 28 using C-reactive protein, RA: rheumatoid arthritis.

* indicates a significant difference at $p < 0.05$

## Drug retention rates and types of JAKis

We also compared the overall drug retention rates between patients with RA treated with pan-JAKi (tofacitinib or baricitinib) and those treated with JAK1i (ubadacitinib filgonicitib) using

**Table 4. Independent risk factors of discontinuation of drug for to AEs inpatients with RA treated with JAKi.**

| | Patients with RA treated with JAKi | | | |
|---|---|---|---|---|
| Variavle | Univariate model | | Multivariable model | |
| | HR (95% CI) | *p*-value | HR (95% CI) | *p*-value |
| Age, >75 yrs or not | 2.09 (1.13–3.86) | 0.02* | 1.44 (0.76–2.74) | 0.27 |
| Disease duration, per 1-yr increase | 0.99 (0.95–1.02) | 0.45 | | |
| Female or not | 0.69 (0.36–1.33) | 0.27 | | |
| RF positive or negative | 1.02 (0.53–1.98) | 0.95 | | |
| ACPA positive or negative | 1.13 (0.56–2.27) | 0.73 | | |
| GC use, yes/no | 1.25 (0.64–2.46) | 0.52 | | |
| MTX use, yes/no | 0.64 (0.33–1.22) | 0.17 | | |
| Coexisting ILD, yes/no | 2.02 (0.96–4.25) | 0.06 | 1.62 (0.76–3.47) | 0.21 |
| Coexisting DM, yes/no | 1.51 (0.72–3.16) | 0.28 | | |
| Pan-JAKi use, yes/no | 6.89 (0.94–50.51) | 0.06 | 5.17 (0.69–38.72) | 0.11 |
| DAS28-CRP >4.1 at JAKi introduction, yes/no | 2.99 (1.61–5.59) | <0.001* | 2.48 (1.31–4.68) | 0.01* |
| No. of previous use of bDMARDs, per drug | 1.07 (0.86–1.33) | 0.53 | | |
| Reduced dose of JAKi, yes/no | 0.87 (0.47–1.60) | 0.65 | | |

JAKi: janus kinase inhibitor, HR: hazard ratio, CI: confidence interval, RF: rheumatoid factor, ACPA: anti-citrullinated peptide antibody, GC: glucocorticoid, MTX: methotrexate, DMARDs: disease-modifying anti-rheumatic drugs, ILD: interstitial lung disease, DM: diabetes mellitus, DAS28-CRP: disease activity score 28 using C-reactive protein, RA: rheumatoid arthritis.

* indicates a significant difference at $p < 0.05$

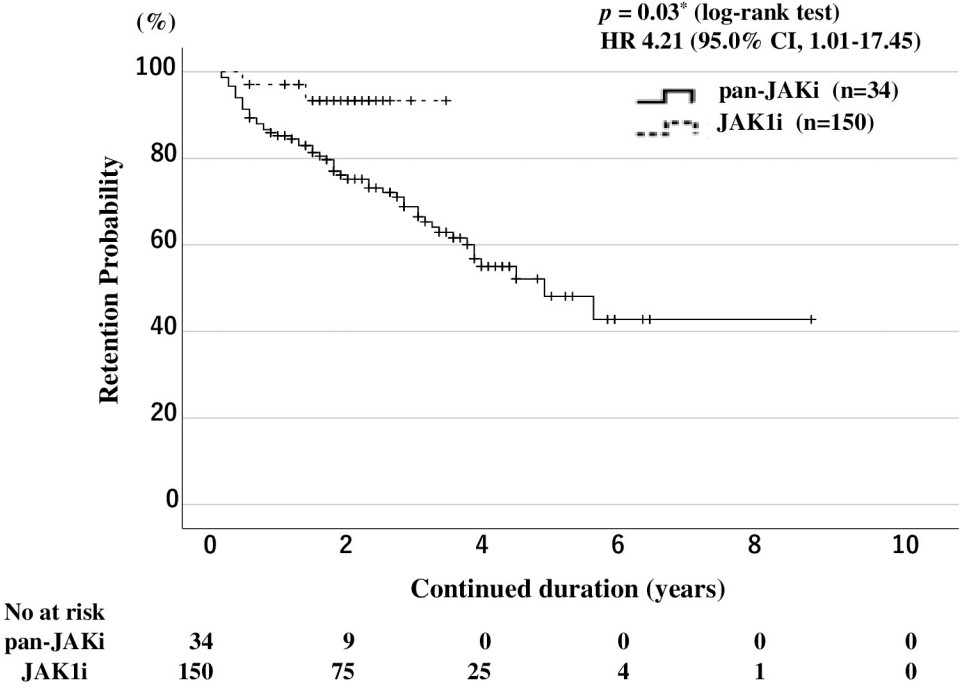

**Fig 2. Overall drug retention rates in pan-JAKi-treated and JAK1i-treated patients.** Kaplan-Meier curves showing the drug retention rates comparing pan-JAKi-treated patients (tofacitinib, baricitinib; n = 150) and JAK1i-treated patients (upadacitinib, filgotinib; n = 34). JAK1i-treated patients have a significantly higher drug continuation rate than pan-JAKi-treated patients. The starting point (0 years) is the date on which the observations began. RA: rheumatoid arthritis; JAKi: Janus kinase inhibitor; No: number.

Kaplan–Meier curves. The overall drug retention rate of JAKis was significantly lower in patients with RA treated with pan-JAKis than in those treated with JAK1is (Fig 2).

## Drug discontinuation rates and RA disease activity in JAKi-treated patients

Finally, we compared the overall drug retention rates between patients with RA with or without high baseline RA disease activity (DAS28-CRP > 4.1) using Kaplan–Meier curves. The overall drug retention rate of JAKis was significantly lower in patients with high RA disease activity (DAS28-CRP >4.1) than in those without high RA disease (Fig 3). However, RA disease activity was differentially associated with lower drug retention rates according to the type of JAKi. Kaplan-Meier curves for the overall drug continuation rate and drug discontinuation rates due to AEs in patients with RA with high disease activity (DAS28-CRP > 4.1) versus those without high disease activity for each JAKi are shown in S1 and S2 Figs, respectively. High RA disease activity was associated with lower drug retention rates in the pan-JAKi (tofacitinib-baricitinib)-treated group (Fig 4A). However, high RA disease activity was not associated with lower drug retention rates in the JAK1i (ubadacitinib-filgonicitib)-treated group (Fig 4B). Furthermore, the drug discontinuations due to AEs were higher in pan-JAKi-treated patients with high RA disease activity (DAS28-CRP >4.1) than in those without high RA disease activity (Fig 5A). Whereas baseline high RA disease activity (DAS28-CRP >4.1) did not affect the drug discontinuations due to AEs in JAK1i-treated RA patients (Fig 5B).

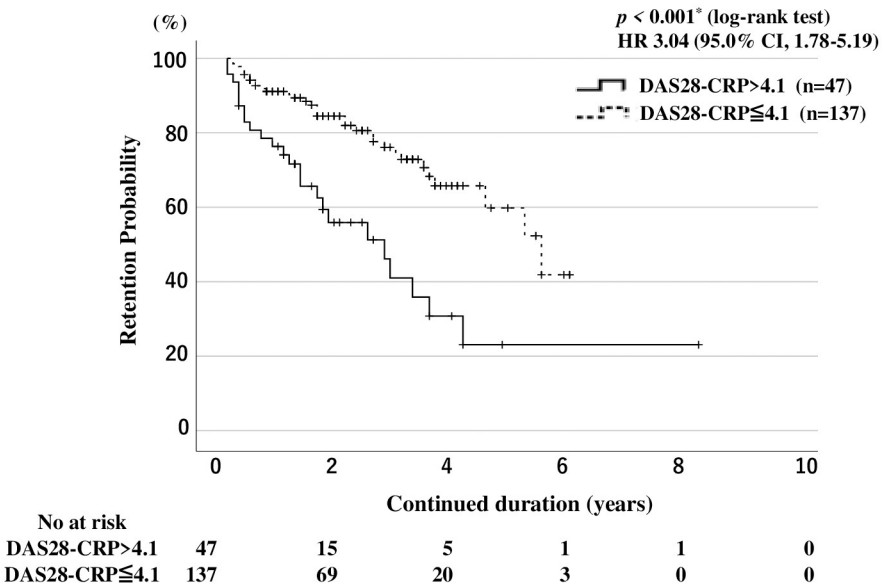

**Fig 3. Drug retention rates in all JAKi-treated patients with and without high disease activity.** Kaplan-Meier curves show the drug retention of JAKi comparing between RA patients with high disease activity and those without high disease activity. Patients having RA with high disease activity have a significantly lower drug continuation rate than those without high disease activity. The starting point (0 years) is the date on which the observations began. RA: rheumatoid arthritis; DAS28-CRP: disease activity score 28 using C-reactive protein, JAKi: Janus kinase inhibitor; No: number.

## Discussion

JAKis offer an alternative therapeutic approach for RA by blocking multiple cytokine cascades implicated in RA pathogenesis [12]. The clinical efficacy of JAKis has been well-established in large randomized controlled trials (RCTs) [13]. In Japan, five JAKis (tofacitinib, baricitinib, peficitinib, upadacitinib, and filgotinib) have been approved for RA treatment [14]. For patients with RA who are refractory to conventional synthetic DMARDs (e.g., MTX), or bDMARDs, JAKis are important treatment options [15]. In real-world settings, JAKis tend to be introduced in patients with RA who are intolerant of bDMARDs failure [16]. Therefore, it is important to investigate the factors that affect the effectiveness and safety of JAKis in patients with refractory RA. In this multicenter cohort study of Japanese patients with RA treated with JAKis, we evaluated drug retention rates of JAKis as first-initiated targeted synthetic DMARDs to assess the safety of JAKis in a real-world population. Our data showed a lower retention rate of pan-JAKis compared with those of JAK1is; however, the statistical significance was weak, likely due to limited power from the small JAK1i-treated group of patients.

Tofacitinib, the first commercially available JAKi, is a pan-JAKi with greater selectivity for JAK1/JAK3 [17]. Baricitinib was developed as a JAK1/JAK2 inhibitor and recently categorized as a pan-JAKi [18]. Filgotinib and upadacitinib selectively target the JAK1 pathway and their rapid and profound efficacy has been demonstrated in clinical trials [19]. Although head-to-head trials using these JAKis have not yet been conducted, the treatment response seems to vary owing to the difference in selectivity of JAKis for each inhibitor [20]. Miyazaki et al. reported real-world data on the efficacy and safety of tofacitinib and baricitinib [21]. By adjusting the baseline data using propensity score-based inverse probability of treatment weighted, they demonstrated that the baricitinib group had a significantly higher rate of clinical disease

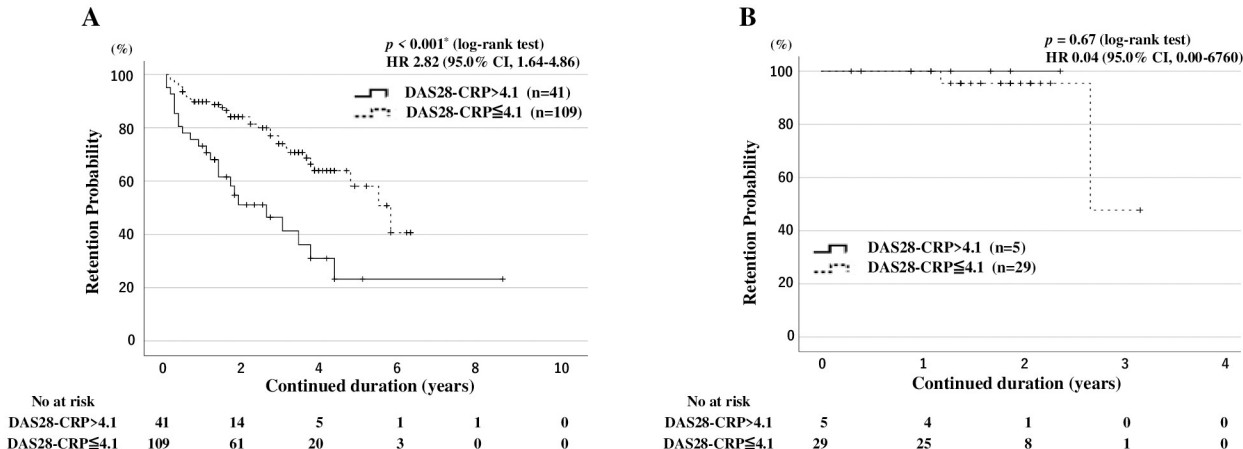

**Fig 4. Drug retention rates in pan-JAKi-treated and JAK1i-treated patients with and without high disease activity.** Kaplan-Meier curves show the drug retention rates in pan-JAKi-treated patients (A; n = 150) and JAK1i-treated patients (B; n = 34) according to the presence of high disease activity. Among pan-JAKi treated patients, patients having RA with high disease activity have significantly lower drug continuation rates than those without. The starting point (0 years) is the date on which the observations began. RA: rheumatoid arthritis; DAS28-CRP: disease activity score 28 using C-reactive protein, JAKi: Janus kinase inhibitor; No: number.

activity index remission at week 24 after drug initiation than did the tofacitinib group [21]. When analyzing the reasons for discontinuation of JAKis, our data showed that AEs were the main reasons for discontinuation rather than a lack of effectiveness. The rate of treatment discontinuation owing to AEs tended to be higher in patients treated with pan-JAKis than in those treated with JAK1is. The significant differences between tofacitinib and JAK1is were small and their clinical significance remained debatable. Further studies on the effectiveness and safety profiles of these JAKis are required.

Interestingly, our results suggested that baseline RA disease activity confers an increased risk of JAKi discontinuation. In multivariable Cox regression analyses, high baseline RA

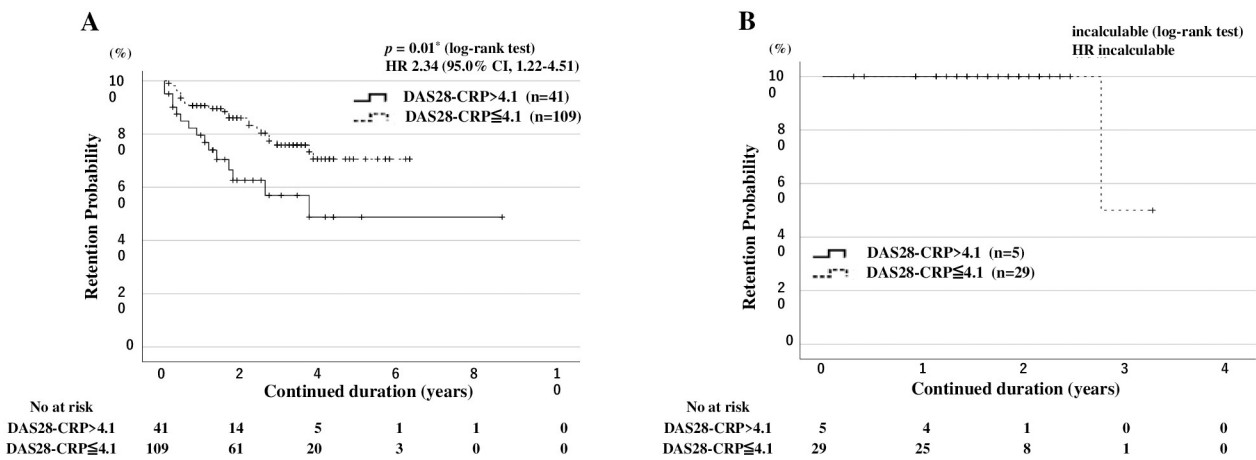

**Fig 5. Drug discontinuation rates due to adverse events in pan-JAKi-treated and JAK1i-treated patients with and without high disease activity.** Kaplan-Meier curves show the drug discontinuation rates due to adverse events in pan-JAKi-treated patients (A; n = 150) and JAK1i-treated patients (B; n = 34) according to the presence of high disease activity. Among pan-JAKi treated patients, Patients having RA with high disease activity had significantly lower drug continuation rates due to adverse events than those without high disease activity t. The starting point (0 years) is the date on which the observations began. RA: rheumatoid arthritis; DAS28-CRP: disease activity score 28 using C-reactive protein, JAKi: Janus kinase inhibitor; No: number.

disease activity (DAS28 >4.1) was identified as a predictive factor of JAKi discontinuation. The Kaplan–Meier results also suggest that high RA disease activity contributes to lower drug retention rates of JAKis owing to AEs. In an observational cohort study, patients with RA receiving JAKis did not demonstrate higher rates of discontinuation owing to AEs than those treated with tumor necrosis factor inhibitors (TNFi) did [22]. In contrast, higher discontinuation rates owing to AEs were observed with tofacitinib than with TNFis in older patients having RA with cardiovascular risk factors [23]. It was reported that RA with lower cerebrovascular attack-free survival rates had highest disease activity levels at baseline [24]. These reports suggest that RA patients with uncontrolled high disease activity may have a higher risk of JAKi discontinuation. Moreover, the ORAL Surveillance study suggested an increased risk of serious AEs with tofacitinib compared with TNFi [8]. Increased risks for these AEs were observed in older tofacitinib-treated patients (> 65 years) or cerebrovascular attack risks indicating potential variations in safety profiles according to the type of JAKi combined with patient's comorbidities [25].

Another interesting finding was that high RA disease activity at the initiation of JAKi treatment was associated with drug discontinuation in patients treated with pan-JAKis (tofacitinib or baricinib) but not in those treated with JAK1is. Inhibition of different components of the JAK family may cause several AEs [26]. JAK2 and JAK3 are expressed in epithelial and hematopoietic cells, and their deficiency results in hematological disorders, including immunodeficiency [27]. Therefore, selective inhibition of JAK1 could minimize the AEs associated with pan-JAK inhibition [28]. After the marketing of the first two pan-JAKis (tofacitinib and baricitinib), more recent research has focused on the development of more selective drugs with the ability to modulate the activity of only one JAK family member (JAK1) to improve the safety profile by minimizing the effects on JAK3 and especially JAK2 [29]. In general, disease activity over time may contribute to the risk of AEs including infections during the treatment for RA [30, 31]. Real-world patients have various comorbidities than those recruited in RCTs. It is possible that RA disease activity affects the drug retention rate of JAKis in RA patients with particular risk factors, which likely depends on JAK selectivity of JAKi. Additional studies are required to determine the relationship between baseline RA disease activity and JAKi retention rates according to the JAK isoform selectivity and RA patient profiles with different risk factors.

The present study had some limitations. First, the observation period was short to draw robust conclusions. In particular, the duration of JAK1i treatment was too short and number of patients was limited. Second, our study lacked detailed analysis, such as treatment responses. Third, the size of the patient cohort was limited, and there was a considerable size difference among the different JAKi-treated patient groups. The judgment and reasons for discontinuation depended on the decisions of each physician without any standardized criteria. Fourth, this was a retrospective study, and the backgrounds of the patients differed among the treatment groups.

## Conclusions

We conducted a cohort study to analyze the drug retention rates of first use differential JAKi treatment. Our results showed that JAK1is (upadacitinib and filgotinib) may have higher drug continuation rates than pan-JAKi (baricitinib and tofacitinib) in real-world patients with RA. The discontinuation rates of pan-JAKi appeared to be higher in patients with high baseline RA disease activity. These novel findings provide new insights into the management of JAKis in clinical practice. Furthermore, large-scale prospective studies are necessary to determine whether baseline high RA disease activity confers drug retention rates on different JAKis.

## Supporting information

**S1 Fig. Drug retention rates in each JAKi-treated patients with and without high disease activity.** The starting point (0 years) is the date on which the observations began. DAS28-CRP: disease activity score 28 using C-reactive protein, JAKi: Janus kinase inhibitor; BARI: baricitinib; TOFA: tofacitinib; UPA: upadacitinib; No: number.
(PPTX)

**S2 Fig. Drug discontinuation rates due to adverse events in each JAKi-treated patients with and without high disease activity.** The starting point (0 years) is the date on which the observations began. DAS28-CRP: disease activity score 28 using C-reactive protein, JAKi: Janus kinase inhibitor; BARI: baricitinib; TOFA: tofacitinib; UPA: upadacitinib; No: number.
(PPTX)

## Acknowledgments

We are grateful to Sachiyo Kanno for her technical assistance in this study.

## Author Contributions

**Conceptualization:** Kenji Saito, Shuhei Yoshida, Kiyoshi Migita.

**Data curation:** Kenji Saito, Shuhei Yoshida, Honoka Ebina, Masayuki Miyata, Eiji Suzuki, Takashi Kanno, Yuya Sumichika, Haruki Matsumoto, Jumpei Temmoku, Yuya Fujita, Naoki Matsuoka, Tomoyuki Asano, Shuzo Sato, Kiyoshi Migita.

**Formal analysis:** Shuhei Yoshida.

**Funding acquisition:** Kiyoshi Migita.

**Investigation:** Kenji Saito, Shuhei Yoshida, Kiyoshi Migita.

**Methodology:** Shuhei Yoshida, Kiyoshi Migita.

**Project administration:** Kenji Saito, Kiyoshi Migita.

**Resources:** Kiyoshi Migita.

**Software:** Kenji Saito, Shuhei Yoshida.

**Supervision:** Masayuki Miyata, Eiji Suzuki, Takashi Kanno, Kiyoshi Migita.

**Validation:** Kenji Saito, Shuhei Yoshida, Masayuki Miyata, Eiji Suzuki, Takashi Kanno, Kiyoshi Migita.

**Visualization:** Shuhei Yoshida, Kiyoshi Migita.

**Writing – original draft:** Kenji Saito, Shuhei Yoshida, Kiyoshi Migita.

**Writing – review & editing:** Kenji Saito, Shuhei Yoshida, Honoka Ebina, Masayuki Miyata, Eiji Suzuki, Takashi Kanno, Yuya Sumichika, Haruki Matsumoto, Jumpei Temmoku, Yuya Fujita, Naoki Matsuoka, Tomoyuki Asano, Shuzo Sato, Kiyoshi Migita.

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
