## [Decision Letter · Decision Letter 0]

15 May 2024

PONE-D-24-09250Real-world comparative study of drug retention of Janus kinase inhibitors in patients with rheumatoid arthritisPLOS ONE

Dear Dr. Migita,

Thank you for submitting your manuscript to PLOS ONE. After careful consideration, we feel that it has merit but does not fully meet PLOS ONE’s publication criteria as it currently stands. Therefore, we invite you to submit a revised version of the manuscript that addresses the points raised during the review process.

Our reviewers found some interests in this manuscript, but also pointed out a number of criticisms that are useful for improving the quality of your manuscript. I ask the authors to fully respond to all comments made by reviewers.

We look forward to receiving your revised manuscript.

Kind regards,

Masataka Kuwana, MD, PhD

Academic Editor

PLOS ONE

- DOI: 10.7573/dic.212595

In your revision ensure you cite all your sources (including your own works), and quote or rephrase any duplicated text outside the methods section. Further consideration is dependent on these concerns being addressed.

“The study was supported by the Japan Grant-in-Aid for Scientific Research (20K08777).”

Reviewers' comments:

Reviewer's Responses to Questions

**Comments to the Author**

1. Is the manuscript technically sound, and do the data support the conclusions?

Reviewer #1: Partly

Reviewer #2: Yes

2. Has the statistical analysis been performed appropriately and rigorously? 

Reviewer #1: Yes

Reviewer #2: Yes

3. Have the authors made all data underlying the findings in their manuscript fully available?

Reviewer #1: Yes

Reviewer #2: Yes

4. Is the manuscript presented in an intelligible fashion and written in standard English?

Reviewer #1: Yes

Reviewer #2: Yes

5. Review Comments to the Author

Reviewer #1: The authors conducted a retrospective study using real-world data to investigate the retention rates of JAK inhibitors and demonstrated that pan-JAK inhibitors and high disease activity were risk factors for discontinuation. Adverse events associated with JAK inhibitors have been a concern in observational studies such as ORAL surveillance, making the authors' report on real-world data valuable.

However, there are several concerns regarding this study, particularly the classification of JAK inhibitors.

Major:

1. The validity of classifying Baricitinib and Tofacitinib as pan-JAK inhibitors needs to be examined. Although both drugs are non-selective for JAK1, the JAK-STAT they inhibit differs. The definition of JAK inhibitor classification (JAK1 inhibitors, pan-JAK inhibitors) should be explicitly stated in the Methods section. While the cited review (reference 17) lists Baricitinib and Tofacitinib as pan-JAK inhibitors, have there been any clinical studies that actually used this classification?

2. Given the differences in overall discontinuation rates and discontinuation rates due to adverse events or insufficient efficacy between Tofacitinib and Baricitinib, the validity of grouping these two drugs together needs to be considered.

3. The appropriateness of grouping Upadacitinib and Filgotinib as JAK1 inhibitors requires further investigation. The number of prescriptions for each drug and the specific drugs associated with discontinuations should be clarified. Table 1 should present the clinical characteristics of Upadacitinib and Filgotinib to assess the validity of grouping them together.

4. Could the risk factors for JAK inhibitor discontinuation be examined based on the presence or absence of each JAK inhibitor in Table 3 and Table 4? Similarly, could Kaplan-Meier curves for each JAK inhibitor be presented in Figure 4 and Figure 5 and included in the Supplementary materials?

5. To evaluate the relationship between adverse events and drugs, the dosage information for each drug should be provided in Table 1. In Japan, Upadacitinib can be administered at 7.5mg, but this information is missing (Page 6, Line 102).

6. In the log-rank tests for Figure 4B and Figure 5B, the event numbers are only 2 and 0, and the group sizes are extremely small, with n=5 and n=29. This raises concerns about low statistical power and instability of the results. Therefore, when interpreting the results and discussing the findings, it is necessary to carefully consider the limitations of the small sample sizes and provide a cautious discussion (Page 15 Lines 226–227, Page 20 Line 316).

7. The conclusions in the abstract (Page 3 Lines 41-43) mention findings that are not presented in the results section of the abstract. Please ensure that all key findings mentioned in the conclusions are first reported in the results section of the abstract. Modify the abstract as needed to maintain consistency between the results and conclusions.

Minor:

1. In Table 3, glucocorticoids are listed as an independent factor for drug discontinuation, but this is not mentioned in the main text (Page 11 Lines 178–180).

2. Should the title of Table 4 be "drug discontinuation of drug for to adverse AEs" rather than all reasons"?

Reviewer #2: Dear authors

This manuscript shown that higher drug retention rates of JAK1 compared to those of pan-JAKis (baricitinib or tofacitinib) in real-world patients with RA. The discontinuation rates of pan-JAKi appeared to be higher in patients with high baseline RA disease activity. The authors honesty noted the limitations of this study include difference of observational period among JAKis, however, there are other concerns on this manuscript.

1. Regarding the multivariate analysis, the variables seem to be too many for the number of cases. This is a simple question, but would you clarify the reason for using those variables?

2. In all JAKis, prescribing at reduced dosages was noticeable, which was most particularly the case with baricitinib. This may also lead to a possible low overall continuation rate. Is there a possibility of significant results depending on the choice of explanatory variables?

3. Despite the limitations of this paper, the retrospective and varying observation period, could you make additional statistical analysis of ROC analysis for drug discontinuation when the patiets have high disease activity at induction of JAKi and pan-JAKi use?

6. PLOS authors have the option to publish the peer review history of their article (what does this mean?). If published, this will include your full peer review and any attached files.

Reviewer #1: No

Reviewer #2: **Yes: **YASUSHI KONDO

---

## [Author Response · Author response to Decision Letter 0]

7 Jun 2024

Dear Reviewer #1, 

Thank you for the time and effort that you have invested in reviewing the manuscript. We appreciate your suggestions and have revised the manuscript accordingly.

Reviewer #1: The authors conducted a retrospective study using real-world data to investigate the retention rates of JAK inhibitors and demonstrated that pan-JAK inhibitors and high disease activity were risk factors for discontinuation. Adverse events associated with JAK inhibitors have been a concern in observational studies such as ORAL surveillance, making the authors' report on real-world data valuable.

However, there are several concerns regarding this study, particularly the classification of JAK inhibitors.

Major:

1. The validity of classifying Baricitinib and Tofacitinib as pan-JAK inhibitors needs to be examined. Although both drugs are non-selective for JAK1, the JAK-STAT they inhibit differs. The definition of JAK inhibitor classification (JAK1 inhibitors, pan-JAK inhibitors) should be explicitly stated in the Methods section. While the cited review (reference 17) lists Baricitinib and Tofacitinib as pan-JAK inhibitors, have there been any clinical studies that actually used this classification?

We appreciate your comment.

As far as we have found, no clinical trials are using the classification of pan-JAK inhibitors and JAK1 selective inhibitors.

However, in recent years, it is common to classify JAK inhibitors in the rheumatology field into pan-JAK inhibitors (ruxolitinib, tofacitinib, baricitinib, peficitinib, delgocitinib and momelotinib) and JAK1 selective inhibitors (upadacitinib and filgotinib) (Ref. PMID: 33950225, DOI: 10.1093/rheumatology/keaa823)(Ref. PMID: 33950230, DOI: 10.1093/rheumatology/keaa895).

As you indicated, we have included definitions of pan-JAK inhibitors and JAK1 selective inhibitors in the Methods section. This important point has been added to the manuscript on page 7, lines 153-154.

“Tofacitinib and baricitinib were defined as pan-JAKis, while upadacitinib and filgotinib as JAK1 selective inhibitors [11].”

2. Given the differences in overall discontinuation rates and discontinuation rates due to adverse events or insufficient efficacy between Tofacitinib and Baricitinib, the validity of grouping these two drugs together needs to be considered.

We appreciate your comment.

We compared the cumulative retention rates of Baricitinib and Tofacitinib. We used the Kaplan-Meier method to compare the cumulative retention rates of Baricitinib and Tofacitinib. There was no statistically significant difference in drug retention rates between the two groups (Log-rank test p=0.242, HR 0.72 (95%CI 0.41-1.25)). Therefore, we believe it is acceptable to lump both drugs together as pan-JAK inhibitors.

3. The appropriateness of grouping Upadacitinib and Filgotinib as JAK1 inhibitors requires further investigation. The number of prescriptions for each drug and the specific drugs associated with discontinuations should be clarified. Table 1 should present the clinical characteristics of Upadacitinib and Filgotinib to assess the validity of grouping them together.

We appreciate your comment.

As you indicated, we have added the clinical characteristics of Upadacitinib (n=22) and Filgotinib (n=12) to Table 1.

In addition, a comparison of the cumulative retention rates of Upadacitinib and Filgotinib was performed. We used the Kaplan-Meier method to compare the cumulative retention rates of Upadacitinib and Filgotinib. There was no statistically significant difference in drug retention rates between the two groups (Log-rank test p=0.43, HR 0.02 (95%CI 0.0-4.1*105)). Therefore, we believe it is acceptable to lump both drugs together as JAK1 selective inhibitors.

4. Could the risk factors for JAK inhibitor discontinuation be examined based on the presence or absence of each JAK inhibitor in Table 3 and Table 4? Similarly, could Kaplan-Meier curves for each JAK inhibitor be presented in Figure 4 and Figure 5 and included in the Supplementary materials?

We appreciate your important comment.

Overall, discontinuation of JAKis occurred in 56 patients, including 32, 22 and 2 treated with Baricitinib, Tofacitinib, and JAK1is, respectively. The two discontinuation events with JAK1 selective inhibitors occurred in patients receiving Upadacitinib. Adverse event discontinuations for each JAK inhibitor were Baricitinib in 23 cases, Tofacitinib in 15 cases, and Upadacitinib in 1 case. Since no discontinuation events occurred in patients who received Filgotinib, it cannot be used as a factor in the Cox regression analysis.

As you indicated, we did a Cox regression analysis using the presence or absence of each JAK inhibitor (Baricitinib, Tofacitinib, and Upadacitinib) as a factor. 

Neither JAK inhibitor was a significant factor in the Cox regression hazard analysis for all discontinuations (Baricitinib: p=0.76, HR 0.92 (95%CI 0.54-1.58); Tofacitinib: p=0.06, HR 1.71 (95%CI 0.99-2.97); Upadacitinib: p=0.18, HR 0.38 (95%CI 0.90-1.55)). When the five factors meeting p<0.2 (age>75, GC use, TOF use, UPA use, DAS28-CRP >4.1 at JAKi introduction) were incorporated into the multivariable Cox regression hazard analysis of all discontinuations, only GC use and DAS28-CRP >4.1 at JAKi introduction were significantly different.

Furthermore, neither JAK inhibitor was a significant factor in the Cox regression hazard analysis of adverse event discontinuation (Baricitinib: p=0.98, HR 1.01 (95% CI 0.54-1.88); Tofacitinib: p=0.09, HR 1.73 (95% CI 0.91-3.27), (p=0.14, HR 0.23 (95%CI 0.03-1.66)), Upadacitinib: p=0.14, HR 0.23 (95%CI 0.03-1.66)). When the five factors meeting p<0.2 (age>75, coexisting ILD, TOF use, UPA use, DAS28-CRP >4.1 at JAKi introduction) were included in the multivariable Cox regression hazard analysis of discontinuation due to adverse events, only DAS28-CRP >4.1 at JAKi introduction was significantly different.

In conclusion, multivariable Cox regression hazard analysis with each JAK inhibitor (Baricitinib, Tofacitinib, and Upadacitinib) as a factor did not reveal any new findings compared to our original analysis. Therefore, we did not add each JAK inhibitor as a factor in Table 4 and Table 5.

Kaplan-Meier curves for each JAK inhibitor (Baricitinib, Tofacitinib, and Upadacitinib) can be generated and are included in the Supplementary Material. This important point has been added to the manuscript on page 15, lines 468-471.

“Kaplan-Meier curves for the overall drug continuation rate and drug discontinuation rates due to AEs in patients with RA with high disease activity (DAS28-CRP > 4.1) versus those without high disease activity for each JAKi are shown in Figures S1 and S2, respectively.”

5. To evaluate the relationship between adverse events and drugs, the dosage information for each drug should be provided in Table 1. In Japan, Upadacitinib can be administered at 7.5mg, but this information is missing (Page 6, Line 102).

We appreciate your important comment.

Reduced doses were administered in 60/106 patients with baricitinib, 19/44 with tofacitinib, 7/22 with upadacitinib, and 5/12 with filgotinib. We have added information on reduced dose of JAKi to Table 1.

As you indicated, information regarding the reduced dose of Upadacitinib was missing. This important point has been added to the manuscript on page 6, lines 138-141.

“The JAKi-treated patients received baricitinib 2 mg (in patients with renal impairment) or 4 mg once daily, tofacitinib 5 mg twice or once daily (in patients with liver impairment), upadacitinib 7.5 mg (in patients with renal impairment) or 15 mg once daily, and filgotinib 100 mg (in patients with renal impairment) or 200 mg once daily.”

6. In the log-rank tests for Figure 4B and Figure 5B, the event numbers are only 2 and 0, and the group sizes are extremely small, with n=5 and n=29. This raises concerns about low statistical power and instability of the results. Therefore, when interpreting the results and discussing the findings, it is necessary to carefully consider the limitations of the small sample sizes and provide a cautious discussion (Page 15 Lines 226–227, Page 20 Line 316).

We appreciate your comment.

As you indicated, all conclusions (and comparisons) are limited due to the small number of patients, short follow-up period, and small number of events in both groups of JAK1is in Figures 4B and 5B. We considered the conclusions too strong and have revised some of the Conclusions. This important point has been added to the manuscript on page 21, lines 632-639.

“Our study showed JAK1is (upadacitinib and filgotinib) may have higher drug continuation rates compared to pan-JAKi (baricitinib and tofacitinib) in real-world patients with RA.”

7. The conclusions in the abstract (Page 3 Lines 41-43) mention findings that are not presented in the results section of the abstract. Please ensure that all key findings mentioned in the conclusions are first reported in the results section of the abstract. Modify the abstract as needed to maintain consistency between the results and conclusions.

We appreciate your comment. We have made some additions to the results section of the abstract as you indicated.

Minor:

1. In Table 3, glucocorticoids are listed as an independent factor for drug discontinuation, but this is not mentioned in the main text (Page 11 Lines 178–180).

We appreciate your important comment.

This important point has been added to the manuscript on page 11, line 365-367.

“Glucocorticoid use, high baseline RA disease activity (DAS28-CRP > 4.1) and pan-JAKi use were identified as independent risk factors for the overall JAKi discontinuation (Table 3).”

2. Should the title of Table 4 be "drug discontinuation of drug for to adverse AEs" rather than all reasons"?

We appreciate your critical comment.

As you indicated, we believe that to AEs is appropriate, not all reasons. We have revised the manuscript.

Dear Reviewer #2, 

Thank you for the time and effort that you have invested in reviewing the manuscript. We appreciate your suggestions and have revised the manuscript accordingly.

Reviewer #2: Dear authors

This manuscript shown that higher drug retention rates of JAK1 compared to those of pan-JAKis (baricitinib or tofacitinib) in real-world patients with RA. The discontinuation rates of pan-JAKi appeared to be higher in patients with high baseline RA disease activity. The authors honesty noted the limitations of this study include difference of observational period among JAKis, however, there are other concerns on this manuscript.

1. Regarding the multivariate analysis, the variables seem to be too many for the number of cases. This is a simple question, but would you clarify the reason for using those variables?

We appreciate your important comment.

We performed univariate and multivariate Cox regression analysis to identify factors associated with the AEs. Variables with p-values <0.2 in the univariate Cox regression analyses were entered in the multivariable Cox regression analysis. Discontinuations for all reasons occurred in 56 patients, and adverse event discontinuations occurred in 39 patients in this study.

Since the number of independent variables that can be analyzed in a multivariate Cox regression analysis is one per 10 events, analysis of up to 5-6 and 3-4 factors is appropriate for risk factors for discontinuation of medication for all reasons and for AEs, respectively. However, in this study, p<0.2 was included as a factor in the multivariate COX regression analysis, so five factors were entered into the analysis of risk factors for AE. As you indicated, we determined that too many factors were entered into the analysis of risk factors for AE. Explanatory variables must be narrowed down to an appropriate number prior to analysis. We chose to include factors that were p<0.2 in the univariate analysis in the multivariate Cox regression analysis for the analysis of risk factors of all discontinuations and AE. In cases where the number of variables that could be entered in the multivariate analysis was limited by the number of outcomes, the more significant variables were adopted.

We described the issue in the materials and methods of the revised manuscript, which can be found on lines 170-174, page 7.

“Variables with p < 0.2 were included in the multivariable Cox regression analysis for the analysis. In cases where the number of variables that could be entered in the multivariate analysis was limited by the number of outcomes, the more significant variables were adopted.”

In addition, we used age, gender, disease duration, rheumatoid factor (RF), anti-citrullinated protein antibody (ACPA), history of bDMARD use, coexisting diabetes mellitus (DM) or pulmonary disease, concomitant medications, and RA disease activity as factors in univariate Cox regression analysis. These factors 【age (Ref. PMID: 20039405, DOI: 10.1002/art.27227), gender (Ref. PMID: 26490106, DOI: 10.1093/rheumatology/kev374), disease duration (Ref. PMID: 30971306, DOI: 10.1186/s13075-019-1880-4), RF and ACPA (Ref. PMID: 26359449, DOI: 10.1136/annrheumdis-2015-207942), history of bDMARD use (Ref. PMID: 25922549, DOI: 10.1093/rheumatology/kev019), coexisting DM or pulmonary disease (Ref. PMID: 24414744, DOI: 10.1007/s00296-014-2945-y), concomitant medications (Ref. PMID: 20039405, DOI: 10.1002/art.27227) (Ref. PMID: 25922549, DOI: 10.1093/rheumatology/kev019), and RA disease activity (Ref. PMID: 25505001, DOI: 10.1093/rheumatology/keu455)】are already considered risk factors for discontinuation of biologics and JAKi. 

We therefore used these variables in our analysis.

2. In all JAKis, prescribing at reduced dosages was noticeable, which was most particularly the case with baricitinib. This may also lead to a possible low overall continuation rate. Is there a possibility of significant results depending on the choice of explanatory variables?

We appreciate your important comment.

Kaplan-Meier analysis of the cumulative incidence of all discontinuations with and without reduced doses of all JAKi and baricitinib alone for rheumatoid arthritis patients showed no significant differences in either outcome (all JAKi; Log-rank p = 0.461, HR 0.82 (95%CI 0.48-1.40)/barcitinib alone; Log-rank p = 0.924, HR 0.97 (95%CI 0.47-1.97)).

In this study, we do not believe that the choice of explanatory variables for the reduced dosage of JAKi will yield significant results.

3. Despite the limitations of this paper, the retrospective and varying observation period, could you make additional statistical analysis of ROC analysis for drug discontinuation when the patiets have high disease activity at induction of JAKi and pan-JAKi use?

We appreciate your comment.

As you indicated, we performed ROC analysis of DAS28-CRP in all JAKi and pan-JAKi of all discontinuations. The cutoff value of DAS28-CRP for all JAKi discontinuations was 4.08 (sensitivity 43.6%, specificity 82.2%, area under the curve 0.61). Furthermore, the cutoff value of DAS28-CRP for all pan-JAKi discontinuations was 3.34 (sensitivity 62.3%, specificity 64.9%, area under the curve 0.61). The results suggest that high RA disease activity may be an important factor for total and adverse event discontinuation of JAKi.

In addition, responses to the journal comments are presented as follows:

1. The manuscript has been formatted according to PLOS ONE style guidelines.

2. We have ensured that all sources are cited in the text and that there is no overlap with DOI: 10.7573/dic.212595.

3. The Role of Funder statement is as follows: The funders had no role in study design, data collection and analysis, decision to publish, or preparation of the manuscript.

4. We confirm that the submission contains all raw data required to replicate the results of our study. Furthermore, we have shared the minimal data set of the study.

Thank you for your consideration. I look forward to hearing from you.

---

## [Decision Letter · Decision Letter 1]

24 Jun 2024

Real-world comparative study of drug retention of Janus kinase inhibitors in patients with rheumatoid arthritis

PONE-D-24-09250R1

Dear Dr. Migita,

We’re pleased to inform you that your manuscript has been judged scientifically suitable for publication and will be formally accepted for publication once it meets all outstanding technical requirements.

Kind regards,

Masataka Kuwana, MD, PhD

Academic Editor

PLOS ONE

Additional Editor Comments (optional):

Reviewers' comments:

Reviewer's Responses to Questions

**Comments to the Author**

1. If the authors have adequately addressed your comments raised in a previous round of review and you feel that this manuscript is now acceptable for publication, you may indicate that here to bypass the “Comments to the Author” section, enter your conflict of interest statement in the “Confidential to Editor” section, and submit your "Accept" recommendation.

Reviewer #1: All comments have been addressed

Reviewer #2: All comments have been addressed

2. Is the manuscript technically sound, and do the data support the conclusions?

Reviewer #1: Yes

Reviewer #2: (No Response)

3. Has the statistical analysis been performed appropriately and rigorously? 

Reviewer #1: Yes

Reviewer #2: (No Response)

4. Have the authors made all data underlying the findings in their manuscript fully available?

Reviewer #1: Yes

Reviewer #2: (No Response)

5. Is the manuscript presented in an intelligible fashion and written in standard English?

Reviewer #1: Yes

Reviewer #2: (No Response)

6. Review Comments to the Author

Reviewer #1: The authors have successfully revised the manuscript according to reviewers' comments. I have no additional comments.

Reviewer #2: (No Response)

7. PLOS authors have the option to publish the peer review history of their article (what does this mean?). If published, this will include your full peer review and any attached files.

Reviewer #1: **Yes: **Shinji Watanabe

Reviewer #2: No

---

## [Editor Report · Acceptance letter]

2 Jul 2024

PONE-D-24-09250R1 

PLOS ONE

Dear Dr. Migita, 

I'm pleased to inform you that your manuscript has been deemed suitable for publication in PLOS ONE. Congratulations! Your manuscript is now being handed over to our production team.

Kind regards, 

on behalf of

Prof. Masataka Kuwana 

Academic Editor

PLOS ONE